# The health and economic burden of respiratory syncytial virus associated hospitalizations in adults

Namrata Prasad[1,2]*, E. Claire Newbern[1]*, Adrian A. Trenholme[3], Mark G. Thompson[4], Colin McArthur[5], Conroy A. Wong[3], Lauren Jelley[1], Nayyereh Aminisani[1,6], Q. Sue Huang[1], Cameron C. Grant[2,7]

1 Institute of Environmental Science and Research, Wellington, New Zealand, 2 Department of Paediatrics: Child & Youth Health, University of Auckland, Auckland, New Zealand, 3 Counties Manukau District Health Board, Auckland, New Zealand, 4 United States Centers for Disease Control and Prevention, Atlanta, GA, United States of America, 5 Department of Critical Care Medicine, Auckland City Hospital, Auckland, New Zealand, 6 Non-Communicable Disease Research Centre, Neyshabur University of Medical Sciences, Neyshabur, Iran, 7 General Paediatrics, Starship Children's Hospital, Auckland, New Zealand

* Namrata.Prasad@esr.cri.nz (NP); Claire.Newbern@esr.cri.nz (CN)

**Data Availability Statement:** Individual level data cannot be shared publicly because of requirements determined by New Zealand Ministry of Health,

## Abstract

### Background

Respiratory syncytial virus (RSV) is increasingly recognized as an important cause of illness in adults; however, data on RSV disease and economic burden in this age group remain limited. We aimed to provide comprehensive estimates of RSV disease burden among adults aged ≥18 years.

### Methods

During 2012–2015, population-based, active surveillance of acute respiratory infection (ARI) hospitalizations enabled estimation of the seasonal incidence of RSV hospitalizations and direct health costs in adults aged ≥18 years in Auckland, New Zealand.

### Results

Of 4,600 ARI hospitalizations tested for RSV, 348 (7.6%) were RSV positive. The median (interquartile range) length of hospital stay for RSV positive patients was 4 (2–6) days. The seasonal incidence rate (IR) of RSV hospitalizations, corrected for non-testing, was 23.6 (95% confidence intervals [CI] 21.0–26.1) per 100,000 adults aged ≥18 years. Hospitalization risk increased with age with the highest incidence among adults aged ≥80 years (IR 190.8 per 100,000, 95% CI 137.6–244.0). Being of Māori or Pacific ethnicity or living in a neighborhood with low socioeconomic status (SES) were independently associated with increased RSV hospitalization rates. We estimate RSV-associated hospitalizations among adults aged ≥18 years to cost on average NZD $4,758 per event.

Health and Disability Ethics Committee. Southern Hemisphere Influenza and Vaccine Effectiveness Research and Surveillance (SHIVERS) data are available from the SHIVERS Data Governance Group (contact via Robert.Press@esr.cri.nz) for researchers who meet the criteria for access to confidential data. The data underlying the results presented in the study are available from (SHIVERS Data Governance Group (contact via Robert. Press@esr.cri.nz).

**Funding:** This work was supported by the US Centers of Disease Control and Prevention funded (cooperative agreement number 1U01IP000480-01) and a Pacific Health Research scholarship by the New Zealand Health Research Council to NP. Support in kind was provided by the New Zealand Ministry of Health.

**Competing interests:** Namrata Prasad, Sue Q Huang, and E Claire Newbern are currently contracted by GlaxoSmithKline for an RSV surveillance project. This does not alter our adherence to PLOS ONE policies on sharing data and materials.

## Conclusions

RSV infection is associated with considerable disease and economic cost in adults. RSV disproportionally affects adult sub-groups defined by age, ethnicity, and neighborhood SES. An effective RSV vaccine or RSV treatment may offer benefits for older adults.

## Introduction

Respiratory syncytial virus (RSV) is well established as a major cause of acute respiratory infections (ARI) in children, but the burden of disease in adults has been less completely studied. Recent studies have reported RSV as a considerable cause of morbidity and mortality in adults [1] sometimes equalling or exceeding rates caused by seasonal influenza [2–4].

While previous studies have been informative, uncertainties about RSV disease burden among adults remain. Most estimates of RSV-associated hospitalization rates are from studies conducted in the USA and appear to vary by location and calendar year. The methods used to estimate RSV disease burden have also varied, with a few studies using active surveillance [2–6], while others have been based on statistical models correlating clinical data with viral activity captured through passive surveillance [7–9]. Moreover, laboratory methods used to confirm RSV have changed over time, with molecular methods having higher sensitivity than previously used serologic and virus isolation techniques [10]. Finally, few studies among adults report RSV hospitalization rates by fine age strata or other demographic characteristics [1].

No licensed RSV vaccine is currently available; however, several adult RSV vaccines candidates are in development [11, 12]. More comprehensive estimates of RSV disease burden in adults will help inform the introduction of such interventions.

In this study, we utilised ARI surveillance data to estimate the incidence of RSV-associated hospitalizations and direct healthcare associated costs among adults aged ≥18 years in Auckland, New Zealand (NZ). We present RSV-associated hospitalization rates estimated by two complementary methods. Our findings provide evidence to inform RSV treatment and preventative strategies in adult populations.

## Methods

In this study, we retrospectively established a cohort of adult residents aged ≥18 years and identified RSV positive ARI hospitalizations within this cohort using data from the Southern Hemisphere Influenza and Vaccine Effectiveness Research and Surveillance (SHIVERS) project (Fig 1). SHIVERS was an active ARI surveillance project conducted in two public hospitals which provide all inpatient services for the population residing in the central, southern, and eastern regions of Auckland [13, 14]. The study area is predominantly urban with an estimated 734,530 adults aged ≥18 years in 2015, of whom 9% are Māori (NZ's indigenous population), 16% Pacific (including ethnic groups from Samoa, Cook Islands, Tonga, Niue, Fiji, Tokelau, Tuvalu, and Kiribati), 23% Asian, and 52% of European or other ethnicities [15]. Ethical approval for the SHIVERS project was obtained from the NZ Ministry of Health, Health and Disability Ethics Committee (NTX/11/11/102).

### Hospital surveillance

From 30 April 2012 to 31 December 2015, research nurses evaluated adults admitted to inpatient wards for suspected ARI. Suspected ARI cases were identified through a

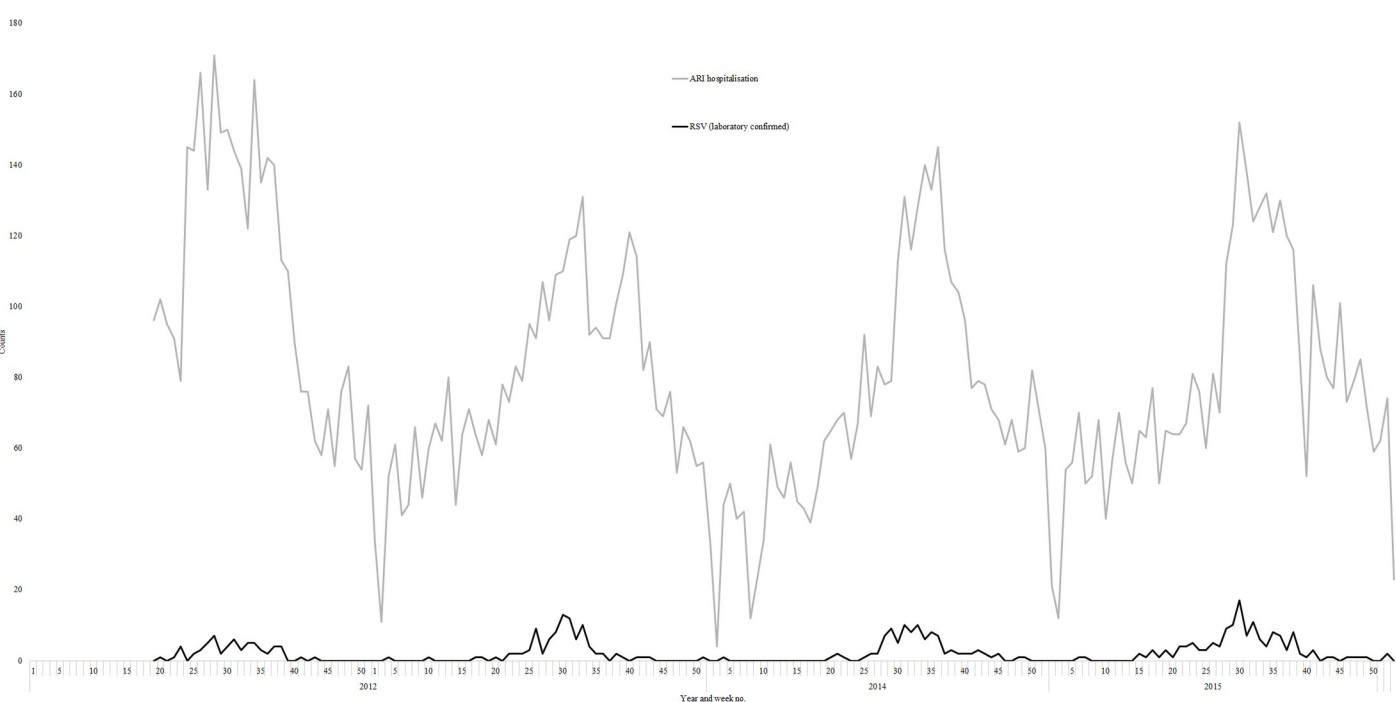

**Fig 1. Weekly counts of acute respiratory infection (ARI) hospitalizations and respiratory syncytial virus (RSV) laboratory-confirmed hospitalizations among adults aged 50 years or older in Auckland, New Zealand, 2012–2015.**

combination of reviewing admission diagnoses and interviewing patients about their presenting signs and symptoms. Among ARI patients, those meeting the WHO severe ARI (SARI) case definition of cough and measured or reported fever, within the last 7 days in 2012, and then within 10 days from 2013 onwards were enrolled. Study nurses obtained consent from eligible patients; completed detailed case report forms on a range of factors including comorbidities, influenza vaccination, antibiotic treatment and clinical outcomes; and collected nasopharyngeal swabs.

To provide an understanding of the respiratory virus hospitalization burden among ARI patients not meeting the SARI definition, in 2013–2015 study nurses enrolled a sample of non-SARI respiratory patients. Sampling in 2013 was during the peak winter period (12 August to 6 October) and included a weekly random selection of four adult inpatients who fitted the non-SARI respiratory definition. In 2014 and 2015, this surveillance was extended to enrol approximately twelve adult non-SARI respiratory patients weekly for a broader winter season (end of April to end of September, week 18–39).

In addition to respiratory virus test results generated by SHIVERS surveillance, hospital laboratories also provided results from clinically ordered tests performed on SARI and non-SARI respiratory patients during the study period. These results were included after validation of the hospital respiratory virus PCR assay. This clinician identified sample was especially valuable in expanding the number of laboratory tested non-SARI patients including in the analysis, since fewer non-SARI patients were systematically enrolled in the SHIVERS study.

Full-year surveillance data showed that RSV follows a well-defined seasonal pattern with 89.5% of RSV positive hospitalizations detected during the broad winter surveillance period (Fig 1). As SHIVERS non-SARI respiratory testing was only conducted during winter seasons, we restricted our analyses to weeks 18–39 of 2012–2015.

## Laboratory methods

Collected specimens were tested for RSV, influenza, rhinovirus, adenovirus, and human metapneumovirus using the United States Centers for Disease Control and Prevention real-time reverse transcription (RT)-PCR protocol [16, 17] at the Institute of Environmental Science and Research, or using the AusDiagnostic PCR protocol and real-time PCR assays at hospital laboratories [18]. A sample of positive specimens were further sub-typed. Further information on performance of hospital assays compared to CDC's real-time RT-PCR as a gold standard are provided in supplementary material (S1 Table).

## Incidence rate denominator data sources

We used national administrative datasets including hospital discharges from the National Minimum Dataset, primary health care enrolments from Primary Health Organizations, births from the National Maternity Collection, deaths from the National Mortality Collection, and specialist visits from the National Non-Admitted Patient Collection [19] to identify and obtain individual level demographic data on adults aged ≥18 years residing in the SHIVERS study area during surveillance periods. These datasets use a unique patient identifier (National Health Index (NHI) numbers) enabling linkage. This population data was linked to SHIVERS data using NHI numbers to identify those with ARI and RSV confirmed hospitalizations.

## Cost estimation

Each inpatient event in the National Minimum Dataset is allocated a Diagnosis Related Group (DRG) code [20]. The DRG coding system categorises hospitalizations into clinically similar events with comparable resource use. These codes, together with information on hospital length of stay and additional interventions such as mechanical ventilation, are used to calculate a cost weight and therefore a cost for each inpatient hospital admission. DRGs are the principal means of reimbursing hospitals for inpatient care in most high-income countries including New Zealand [21]. We calculated hospitalization costs by multiplying each cost weight provided in the NMDS with the NZ fixed cost multiplier applicable for the 2017/18 financial year [20].

## Statistical analysis

Chi-square tests were used to test associations between categorical variables and Student's t-tests for continuous variables. Among hospitalized cases, logistic regression with adjustment for age and ethnicity was used to test for association of comorbidities, influenza vaccination, and antibiotic treatments with RSV positivity.

For incidence calculations, if a RSV positive patient was transferred to another hospital and/or had multiple RSV positive hospitalizations within 14 days of discharge from their first hospitalization, they were considered a singular illness episode. Incidence rates were calculated by dividing the number of RSV-associated ARI hospitalizations (singular episodes) by the number of adults residing in the study area. Confidence intervals for incidence rates and rate ratios were based on the Poisson distribution. Rates were stratified by Auckland sub-region, age, sex, ethnicity, and SES as they are considered key modifiers of ARI hospitalization risk. SES was quantified using New Zealand's small area measure of neighborhood deprivation derived from the national census (NZ Deprivation Index 2013) [22]. This SES measure was used to divide the study sample into SES quintiles (quintile 1 –least deprived to quintile 5 – most deprived).

In New Zealand, Māori and Pacific peoples have lower reported life expectancy than other ethnic groups [23], and are also overrepresented in the lower SES groups [24]. To control for confounding and evaluate the independent effects of these factors, RSV hospitalization rates and rate ratios for age group, ethnicity, and SES were adjusted for each other.

The correction of incidence rate calculations for non-testing for RSV among ARI patients was done using two methods. First, we multiplied the proportion of those tested who were positive for RSV by the number of non-tested ARI patients within strata formed by study year, sub-region, age group, SES quintile, and ethnicity (multiplier method). Second, we corrected test results using the multivariate imputation by chained equations (MICE) method [25], which has been shown to yield unbiased estimates when accounting for missing outcome data [26]. For imputation, we verified that non-tested patients were missing at random, using a Missing Completely at Random Test [27], before creating 30 imputed datasets of RSV results with age, ethnicity, SES, SARI case definition, sex, week of hospitalization, and specimen type (clinician-ordered versus SHIVERS systematic sampling) included as predictors of missingness. To ensure correction of non-testing was only done for singular RSV illness episodes, patients with two or more untested ARI hospitalizations within 14 days of each other were considered a singular episode. All analyses were performed using Stata 14 (College Station, TX:StataCorp LP).

## Results

### Study population

Based on our linked data methodology, we identified an annual average of 731,204 adults aged ≥18 years residing in the study area during 2012–2015. This was similar to the 2015 Statistics NZ population estimates (734,530) [15].

### Hospitalized patients

The temporal distribution of RSV hospitalizations among adults demonstrated a consistent peak during the New Zealand winter (May-September) (Fig 2). Over the four winter seasons, there were 8,776 ARI hospitalizations among adults aged ≥18 years, including 3,804 (43.4%) SARI events and 4,972 (56.7%) non-SARI events. Of ARI hospitalizations, 4,600 (52.4%) were tested for RSV (Fig 2). Of the 348 unique RSV-associated hospitalization episodes identified, there were 226 (7.4%) from the 3,046 SARI tested hospitalizations and 122 (7.9%) from the 1,554 non-SARI tested hospitalizations.

Non-tested cases were older and a smaller proportion of them were of Asian ethnicity (S2 Table). The patients tested through the SHIVERS study and those tested based on clinician's orders differed by age, SES, ethnicity, SARI case definition, and length of hospital stay (S3 Table).

Among RSV positive samples, 103 (29.6%) were subtyped of which 54 (52.4%) were RSV-A and 49 (47.5%) RSV-B. There were 47 (18.0%) co-detections of RSV with other viruses. Of the 261 RSV positive samples tested for other respiratory viruses, 26 (10.0%) were influenza co-detections, 13 (5.0%) rhinovirus, 6 (2.3%) were adenovirus, and 3 (1.1%) were human metapneumovirus co-detections.

The median (interquartile range [IQR]) hospital length of stay in days among all ARI hospitalizations was 3 (2–6). Of ARI hospitalizations, 179 (2.0%) were admitted to the intensive care unit (ICU), 244 (2.8%) died during their hospital stay, and 312 (3.6%) died within 30 days of discharge. The median (interquartile range [IQR]) hospital length of stay for RSV positive adults was 3 (2–6) days. Among RSV positive cases, 4 (1.1%) died during hospitalization and 11 (3.2%) died within 30 days of hospital discharge.

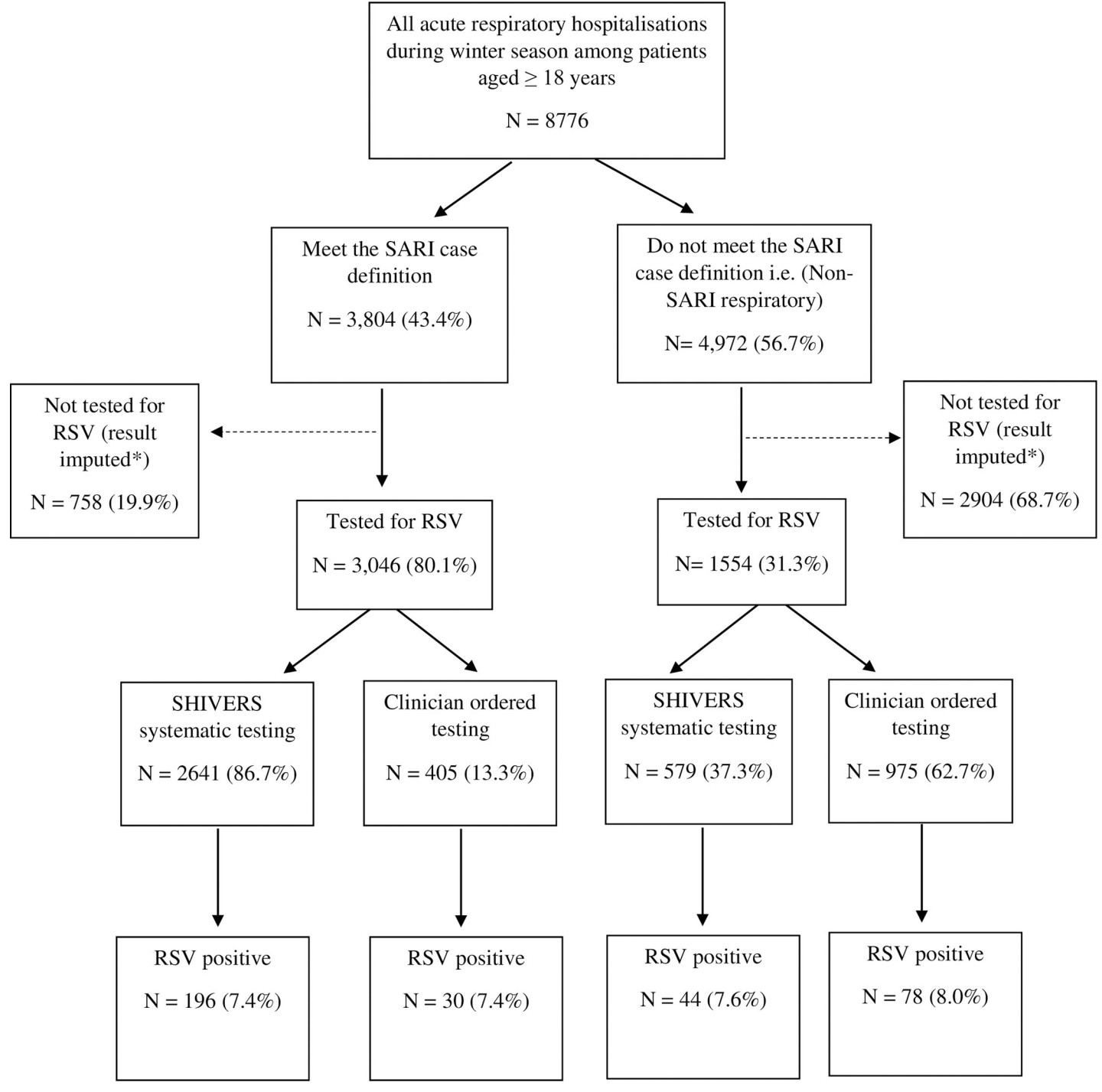

**Fig 2. Flowchart detailing retrospective cohort of adults aged ≥50 years in Auckland, New Zealand in 2012–2015 and number of acute respiratory infection (ARI) and respiratory syncytial virus (RSV)-tested hospitalizations.** *For incidence rate calculations, correction of non-testing among ARI patients was done using two methods; first by multiplying the proportion positive for RSV in each demographic strata to non-tested ARI patients in each group; and second by using the multivariate imputation by chained equations (MICE) method of imputation in STATA [25].

Hospital length of stay did not differ by RSV positivity (p-value = 0.590). The proportion of ARI hospitalizations requiring ICU admission also did not differ significantly by RSV positivity (RSV positive = 8/348 (2.3%) vs RSV negative = 132/4252 (3.1%); p-value = 0.400).

Of 8,776 ARI hospitalizations, 3,421 (39.0%) had additional information on comorbidities, current influenza vaccination status, and antibiotic treatment following hospital admission (Fig 3). Of these cases, 1,740 (50.9%) were regular smokers, 1,970 (57.6%) patients had a current influenza vaccination, and 3,035 (88.7%) received antibiotics during the hospital admission. In terms of comorbidities, 1,523 (44.5%) cases had a chronic lung disease, and 1,284 (37.5%) had a cardiovascular condition. No factors were significantly associated with RSV positivity following adjustment for age and ethnicity.

### Seasonal incidence of RSV confirmed hospitalizations

The seasonal incidence of RSV hospitalization without accounting for non-tested adults was 11.9 (95% confidence interval [CI] 10.7–13.2) per 100,000 adults (Table 1). Following correction for non-testing, the incidence of RSV-associated ARI hospitalizations was 22.7 (95% CI: 21.0–24.5) per 100,000 persons using the multiplier method and 23.6 (95% CI 21.0–26.1) per 100,000 adults using multiple imputation (Table 1). Rates were not significantly different based on the method used for correction, however multiple imputation estimates had slightly wider confidence intervals due to the incorporation of uncertainty inherent in imputation. Subsequent presentation of incidence estimates used data derived from multiple imputation.

Following adjustment for age group, SES, and ethnicity; incidence increased by age, with adults aged 80 years or older being approximately 30 times as likely to have a RSV-associated hospitalization compared to those aged 18–49 years (Rate ratio [RR] 31.3, 95% CI 22.3–44.0; Fig 4). Adults from SES quintiles 3–5 were at higher risk of a RSV-associated hospitalization compared to adults in the least-disadvantaged SES group (quintile 1), following adjustment for age group and ethnicity (Fig 4). Similarly, RSV hospitalization rates adjusted for age group and SES were higher in Māori (RR 2.8, 95% CI 2.0–4.0) and Pacific adults (RR 3.5, 95% CI 2.6–4.7) compared to those of European or other ethnicities (Fig 4).

### Direct health-care costs

Based on the DRG costing methodology, the median (IQR) cost per RSV hospitalization among adults was NZD $3,723.84 ($2500.44-$5,028.44) while among non-RSV hospitalizations the median (IQR) cost per hospitalization was $3,950.71 ($2423.67-$5028.44). Hospitalization costs were not significantly different by RSV positivity in adults (p-value = 0.254). After accounting for non-testing, the annual direct health-care cost of RSV-confirmed hospitalizations in Auckland was NZD $818,398.78 or an average NZD $4,758.13 per hospitalization. This is equivalent to USD $525,137.85 or an average USD $3,053.13 per hospitalization, using November 2019 currency exchange rates.

## Discussion

We present estimates of RSV hospitalizations rates among adults aged 18 years or more from population based ARI surveillance. We show RSV to be associated with approximately 8% of ARI hospitalizations during winter seasons in this age group and to cost an average NZD $4,758 per episode. Increasing age, Māori or Pacific ethnicity and low neighborhood SES were all independently associated with an increased risk of RSV-associated hospitalization in this population.

Of year-round RSV-associated SARI hospitalizations, 90% of RSV occurred during the winter season suggesting that seasonal incidence approximates annual incidence. We observed

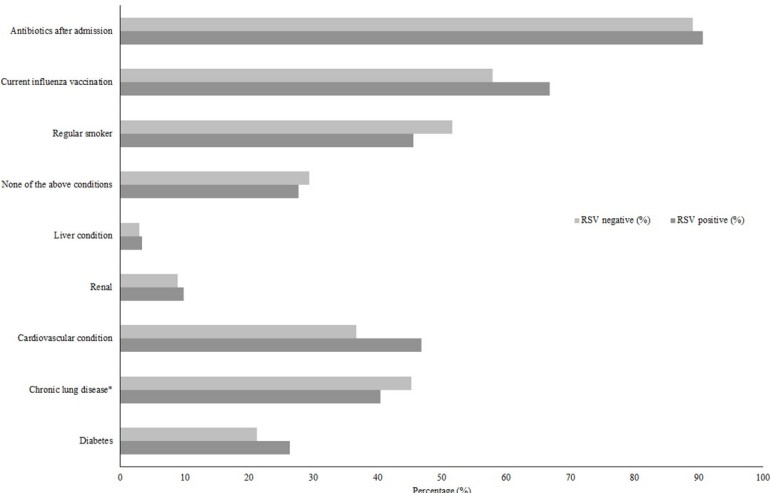

**Fig 3. Risk factors among acute respiratory infection (ARI) hospitalizations in adults aged 18 years or more in Auckland, New Zealand, 2012–2015 by respiratory syncytial virus (RSV) result.** Among 3,421 patients with complete information on comorbidities, influenza vaccination status, and antibiotic treatment. There was no significant difference in risk factors by RSV positivity status after adjusting for age and ethnicity. * Includes asthma, COPD, and bronchiectasis.

disparities in RSV-associated hospitalization rates by age, ethnicity and SES. Such disparities have been reported for RSV-associated hospitalization rates in children [28], but to our knowledge this is the first study to demonstrate such inequalities among adults. Ethnic differences in respiratory infectious diseases are thought primarily to be due to social and economic factors, as ethnic groups with increased risk of infection and disease are also overrepresented in lower socio-economic groups. However in our study we found both ethnicity and SES to have independent effects on RSV-associated hospitalization risk. A potential explanation is that our neighborhood-level measure of SES may not accurately capture individual SES. Consequently, social and economic factors associated with RSV disease risk such as smoking, housing density, presence of comorbidities, health care seeking behaviour, and poor nutrition are potentially being better captured by ethnicity. As our investigation is exploratory, further investigation is warranted to guide policy development. Nevertheless our findings highlight the value of assessing RSV related health disparities among different ethnic populations in other countries.

Our estimates of RSV positivity among adult ARI hospitalizations is comparable to one US study using active ARI surveillance and laboratory confirmation among hospitalized adults aged ≥65 years [2]. When comparing our estimates to studies reporting RSV hospitalization rates using hospital surveillance [3, 4], we found our rates (23.6 per 100,000 adults aged ≥18 years and 99.2 per 100,000 adults aged ≥65 years) to be lower than those reported in these studies (55 per 100,000 adults aged ≥18 years and 189–254 per 100,000 adults aged ≥65 years). One possible reason for differences could be the approach used to extrapolate RSV positivity in untested patients. In our study, all ARI patients were actively identified by study nurses during a broad winter season, and RSV positivity for those without study testing was estimated while accounting for demographic and clinical differences using a traditional multiplier method and a multiple imputation method. In the US studies, RSV hospitalization rates were estimated by multiplying the proportion of enrolled patients positive for RSV by the total number of residents with an ICD-9 classified ARI hospitalization during influenza seasons [3, 4]. Additionally, one of the studies took place during a single year when there was a novel

**Table 1. Seasonal incidence rates (IR) of acute respiratory infection (ARI) and laboratory-confirmed respiratory syncytial virus (RSV)-associated ARI hospitalizations among adults 18 years and older by year, Auckland sub-region, demographic factors in Auckland, New Zealand, 2012–2015.**

| | | | ARI hospitalizations | | | Incidence rates (IR) per 100,000 adults | | |
| --- | --- | --- | --- | --- | --- | --- | --- | --- |
| | | | | | | Crude—Not corrected for non-testing | Corrected for non-testing* | |
| | | | | | | | Multiplier method | Multiple imputation |
| | No. of adults | Total | No. tested for RSV | No. confirmed with RSV | (%) of tested | IR (95% CI) | IR (95% CI) | IR (95% CI) |
| Total | 2924815 | 8776 | 4600 | 348 | (7.6) | 11.9 (10.7–13.2) | 22.7 (21.0–24.5) | 23.6 (21.0–26.1) |
| Season | | | | | | | | |
| 2012 | 708332 | 2637 | 951 | 59 | (1.3) | 8.2 (6.1–10.3) | 23.2 (19.8–26.9) | 25.5 (18.2–32.9) |
| 2013 | 726127 | 2036 | 932 | 83 | (1.8) | 11.3 (8.8–13.7) | 24.9 (21.4–28.8) | 29.2 (23.2–35.2) |
| 2014 | 736724 | 2023 | 1315 | 86 | (1.9) | 11.7 (9.2–14.1) | 17.9 (15.0–21.3) | 19.1 (15.2–23.0) |
| 2015 | 753632 | 2080 | 1402 | 120 | (2.6) | 15.9 (13.1–18.8) | 23.6 (20.3–27.4) | 25.1 (20.4–29.7) |
| Region | | | | | | | | |
| Central Auckland | 1435862 | 4061 | 2063 | 158 | (3.4) | 11.0 (9.3–12.8) | 21.6 (19.3–24.2) | 22.7 (19.3–26.0) |
| South East Auckland | 1488953 | 4715 | 2537 | 190 | (4.1) | 12.8 (10.9–14.7) | 23.7 (21.3–26.3) | 24.4 (21.1–27.8) |
| Age Group (years) | | | | | | | | |
| 18–49 | 1838952 | 2117 | 1367 | 64 | (1.4) | 3.5 (2.6–4.3) | 5.4 (4.4–6.6) | 5.9 (4.3–7.5) |
| 50–64 | 658708 | 1974 | 1129 | 93 | (2.0) | 14.2 (11.3–17.1) | 24.8 (21.2–28.9) | 24.2 (18.2–30.2) |
| 65–79 | 327234 | 2677 | 1308 | 118 | (2.6) | 36.4 (29.6–43.2) | 74.0 (64.9–83.8) | 72.9 (57.4–88.3) |
| ≥65 | 427155 | 4685 | 2104 | 191 | (4.2) | 44.7 (38.6–51.5) | 99.5 (90.3–10.9) | 99.2 (82.4–115.9) |
| ≥80 | 99921 | 2008 | 796 | 73 | (1.6) | 74.5 (57.4–91.5) | 184.1 (158.5–212.8) | 190.8 (137.6–244.0) |
| Sex | | | | | | | | |
| F | 1543948 | 4682 | 2528 | 198 | (4.3) | 12.9 (11.1–14.7) | 23.8 (21.4–26.3) | 23.9 (20.7–27.1) |
| M | 1380867 | 4094 | 2072 | 150 | (3.3) | 10.9 (9.1–12.7) | 21.4 (19.1–24.0) | 23.3 (19.8–26.7) |
| SES (unadjusted) † | | | | | | | | |
| 1 | 525511 | 931 | 474 | 35 | (0.8) | 6.7 (4.5–8.9) | 13.1 (10.2–16.6) | 13.9 (9.0–18.7) |
| 2 | 566525 | 1216 | 624 | 49 | (1.1) | 8.5 (6.1–10.9) | 16.7 (13.6–20.5) | 17.3 (12.7–22.0) |
| 3 | 512159 | 1335 | 676 | 52 | (1.1) | 10.6 (7.8–13.4) | 20.1 (16.4–24.4) | 22.1 (16.5–27.7) |
| 4 | 383620 | 1258 | 639 | 57 | (1.2) | 13.6 (9.9–17.3) | 29.2 (24.0–35.1) | 28.1 (20.2–36.0) |
| 5 | 937001 | 4036 | 2187 | 155 | (3.4) | 17.1 (14.3–19.8) | 30.5 (27.1–34.3) | 31.7 (26.9–36.6) |
| Ethnicity (unadjusted) | | | | | | | | |
| Māori | 249970 | 1585 | 826 | 43 | (0.9) | 17.3 (12.2–22.5) | 33.3 (26.5–41.2) | 34.3 (24.7–43.9) |
| Pacific | 482596 | 2552 | 1452 | 121 | (2.6) | 25.2 (20.6–29.9) | 44.1 (38.4–50.5) | 45.3 (37.2–53.4) |
| Asian | 684059 | 811 | 440 | 32 | (0.7) | 4.7 (3.1–6.3) | 8.6 (6.6–11.1) | 9.2 (6.0–12.4) |
| European/Other | 1508190 | 3828 | 1882 | 152 | (3.3) | 10.1 (8.5–11.7) | 20.5 (18.3–22.9) | 21.4 (17.8–25.0) |

* Correction for non-testing among ARI patients was done using two methods; first by multiplying the proportion positive for RSV in each demographic strata to non-tested ARI patients identified in each group; and second by using the multivariate imputation by chained equations (MICE) method of imputation [25].

† SES quantified using a small area level measure of neighborhood deprivation derived from the national census (NZDep2013) [22].

influenza A H1NI pandemic [3], thus rates estimated that year may not be representative of all years

The relative burden of influenza compared to that due to RSV also differed in comparison to these earlier studies. In the studies by Falsey and Widmer et al, the proportion of ARI hospitalizations in adults positive for influenza were found to be similar to those for RSV. In our study, the proportion of hospitalized adults with influenza (approximately 24%) and the

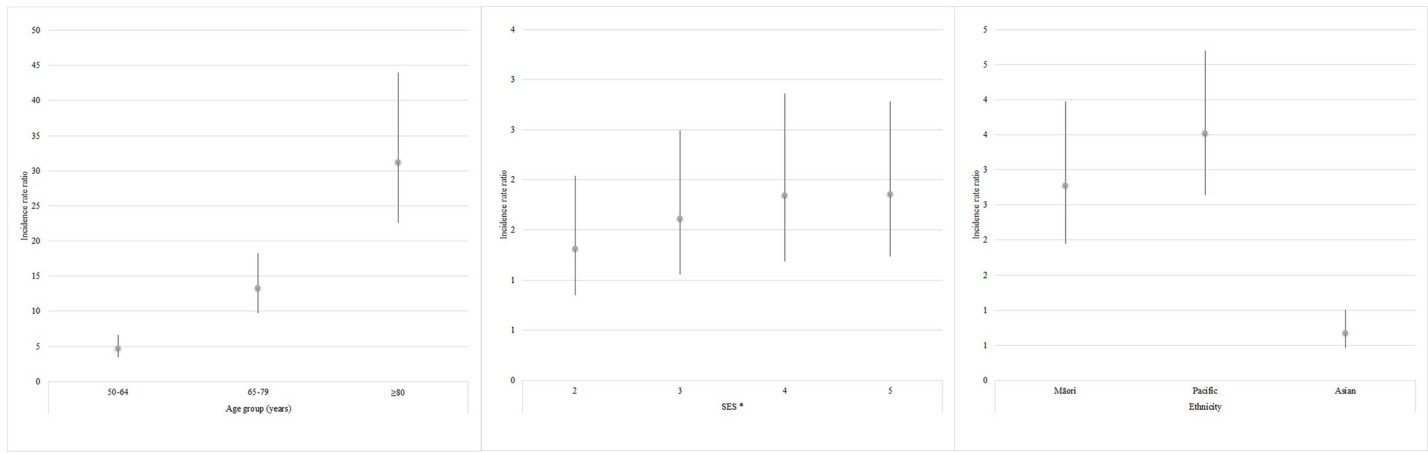

**Fig 4. Adjusted incidence rate ratios for age group (referent 18–49 years old), socio-economic status (referent–quintile 1), and ethnicity (referent–European/other) of respiratory syncytial virus (RSV) associated acute respiratory infection (ARI) hospitalizations among adults 50 years or older in Auckland, New Zealand, 2012–2015.** *Incidence rate ratios for age, socio-economic status and ethnicity have been adjusted for each other. † SES quantified using a small area level measure of neighborhood deprivation derived from the national census (NZDep2013) with SES 1 as least deprived and SES 5 as most deprived [22].

influenza-related hospitalization rate (approximately 72 per 100,000) [29–31] were higher than those for RSV, despite influenza vaccination coverage in adult populations in NZ being similar to those of the US study locations. It is possible that these differences are due to the effects of different circulating influenza strains. In our study, influenza H3N2 was the predominant strain every year except 2014, and has been shown to result in more severe illness than RSV [32], whereas in the US studies, influenza H1N1 was generally the predominant strain.

The RSV-associated hospitalization rates among adults aged ≥65 years (99 per 100,000 adults) reported in this study were similar to estimates reported by studies using indirect statistical modelling. Zhou et al modelled hospitalization and viral surveillance data in 13 US states from 1993–1994 through 2007–2008 and estimated a RSV hospitalization rate of 86 per 100,000 adults aged ≥65 years [9]. Similarly, Fleming et al modelled viral surveillance and hospitalization data from the UK from 1995–2009 and estimated a RSV hospitalization rate of 156 per 100,000 adults aged ≥65 years [7]. Finally, similar to our findings, Zhou et al estimated hospitalization rates due to influenza in adults to be considerably higher than that for RSV, particularly in years dominated by influenza H3N2 [9].

When carrying out analyses among hospitalized cases, we did not find any assessed comorbidity to be significantly associated with RSV positivity following adjustment for age and ethnicity. This lack of association is likely due to the higher risk of hospitalization for non-RSV illnesses in adults with comorbidities i.e. selection bias that occurs when both exposure and outcome are associated with hospitalization. Thus in order to accurately measure the effect of comorbidity on RSV disease risk, population level comorbidity data should be used to estimate RSV-associated hospitalization rates in specific co-morbidity strata. Such estimates will be valuable in identifying adult groups who are at particularly high-risk from RSV-associated disease and priority groups for future RSV vaccines and treatments. Unfortunately, we did not have population-level comorbidity data to carry out such analyses in the current study, however we intend to do so in future work.

A strength of this study is its use of ARI surveillance across multiple seasons linked with individual-level population data, to estimate RSV hospitalization rates by demographic characteristics important for New Zealand. Nonetheless our study had important limitations. First of

which was the lack of RSV testing for all ARI patients and differences in testing by SARI case definition. However, our estimates were similar using two different correction methods for non-testing. Additionally, by using multiple imputation we were able to provide more realistic measures of uncertainty for our rate estimates. Secondly, our estimations of cost are based on direct health care associated cost and do not account for indirect costs associated with loss of work hours and out-of-pocket expenses. Finally, the lack of systematic bacterial and viral co-infection data prevented detailed assessments of the causal role of RSV infection in adult ARIs. However a recent meta-analysis suggests strong evidence for this association, showing that among individuals with RSV-ARI, ARI is causally attributable to RSV in about 88% of cases [33].

## Conclusion

In a setting outside of the United States, where most studies assessing the burden of RSV in older have been conducted, we confirm that RSV has a considerable hospitalization burden and associated economic cost in adults. In our study from 2012–2015, RSV burden was less than that for influenza. RSV disproportionally affects adults by age. Being of Indigenous Māori or Pacific ethnicity or from a low socio-economic area pose independent risks for RSV-associated hospitalization. An effective RSV vaccine or treatment may offer benefits for older adults.

## Supporting information

**S1 Table. Performance of hospital assays (clinician ordered tests) in detecting different respiratory viruses 2012–2016.**
(DOCX)

**S2 Table. Comparison of tested and non-tested ARI hospitalizations stratified by SARI case definition among adults aged 18 or older in Auckland, New Zealand 2012–2015.**
(DOCX)

**S3 Table. Comparison of SHIVERS systematic and clinical ordered testing among ARI hospitalizations in adults aged 18 or older in Auckland, New Zealand 2012–2015.**
(DOCX)

## Acknowledgments

The authors appreciate the contributions of (1) research nurses at Auckland District Health Board (ADHB): Kathryn Haven, Bhamita Chand, Pamela Muponisi, Debbie Aley, Claire Sherring, Miriam Rea, Judith Barry, Tracey Bushell, Julianne Brewer, Catherine McClymont; (2) research nurses at Counties Manukau District Health Board (CMDHB): Shirley Laurence, Shona Chamberlin, Reniza Ongcoy, Kirstin Davey, Emilina Jasmat, Maree Dickson, Annette Western, Olive Lai, Sheila Fowlie, Faasoa Aupa'au, Louise Robertson; (3) researchers at the WHO National Influenza Centre, Institute of Environmental Science and Research (ESR): J. Bocacao, W. Gunn, J. Ralston, P. Kawakami, S. Walker, R. Madge, A. des Barres, Angela Todd; (4) researchers at the ADHB Laboratory (Fahimeh Rahnama); (5) the CMDHB Laboratory: Helen Qiao, Fifi Tse, Mahtab Zibaei, Tirzah Korrapadu, Louise Optland, Cecilia Dela Cruz); (6) Labtests Laboratory in Auckland; (7) researchers involved in the Southern Hemisphere Influenza and Vaccine Effectiveness Research and Surveillance (SHIVERS) project: Tiffany Walker, Ben Waite, Tim Wood, Diane Gross, Jazmin Duque, Sally Roberts, Susan Taylor, Nikki Turner, Richard Webby, Paul Thomas, Ange Bissielo, Don Bandaranayake, and Sarah

Radke; (8) US Centers of Disease Control and Prevention (CDC) reviewers: David Shay, Jerome Tokars, and David Bell.

## Author Contributions

**Conceptualization:** Namrata Prasad, E. Claire Newbern, Adrian A. Trenholme, Nayyereh Aminisani, Q. Sue Huang, Cameron C. Grant.

**Data curation:** Namrata Prasad, Adrian A. Trenholme, Mark G. Thompson, Colin McArthur, Conroy A. Wong, Lauren Jelley, Q. Sue Huang, Cameron C. Grant.

**Formal analysis:** Namrata Prasad, E. Claire Newbern.

**Funding acquisition:** Q. Sue Huang.

**Investigation:** E. Claire Newbern, Lauren Jelley, Q. Sue Huang.

**Project administration:** E. Claire Newbern, Mark G. Thompson, Q. Sue Huang.

**Supervision:** E. Claire Newbern, Adrian A. Trenholme, Mark G. Thompson, Cameron C. Grant.

**Writing – original draft:** Namrata Prasad.

**Writing – review & editing:** Namrata Prasad, E. Claire Newbern, Adrian A. Trenholme, Mark G. Thompson, Colin McArthur, Conroy A. Wong, Lauren Jelley, Nayyereh Aminisani, Q. Sue Huang, Cameron C. Grant.

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
