## [Decision Letter · Decision Letter 0]

6 Apr 2020

PONE-D-19-32957

The health and economic burden of respiratory syncytial virus associated hospitalizations in adults

PLOS ONE

Dear Ms. Prasad,

Thank you for submitting your manuscript to PLOS ONE. After careful consideration, we feel that it has merit but does not fully meet PLOS ONE’s publication criteria as it currently stands. Therefore, we invite you to submit a revised version of the manuscript that addresses the points raised during the review process.

ACADEMIC EDITOR: 

Both reviewers found merit in your work and the quality of the manuscript - each has specific issues that require detailed review at this stage. Reviewer 2 has made an important comment on associations founded on ethnicity which is an increasingly important issue globally.  

We would appreciate receiving your revised manuscript by May 21 2020 11:59PM. To enhance the reproducibility of your results, we recommend that if applicable you deposit your laboratory protocols in protocols.io, where a protocol can be assigned its own identifier (DOI) such that it can be cited independently in the future. For instructions see: http://journals.plos.org/plosone/s/submission-guidelines#loc-laboratory-protocols

We look forward to receiving your revised manuscript.

Kind regards,

Richard van Zyl-Smit

Academic Editor

PLOS ONE

Journal Requirements:

"NP, SH, and ECN are currently contracted by GlaxoSmithKline for an RSV surveillance project."

Additional Editor Comments (if provided):

Thank you for your submission - both reviewers have comments that require some thought and review.

Reviewers' comments:

Reviewer's Responses to Questions

**Comments to the Author**

1. Is the manuscript technically sound, and do the data support the conclusions?

Reviewer #1: Yes

Reviewer #2: Partly

2. Has the statistical analysis been performed appropriately and rigorously? 

Reviewer #1: N/A

Reviewer #2: No

3. Have the authors made all data underlying the findings in their manuscript fully available?

Reviewer #1: Yes

Reviewer #2: Yes

4. Is the manuscript presented in an intelligible fashion and written in standard English?

Reviewer #1: Yes

Reviewer #2: Yes

5. Review Comments to the Author

Reviewer #1: The authors retrospectively review a cohort of Auckland, New Zeland adult persons aged ≥18 years and identified RSV positive among ARI hospitalizations, from 2012 to 2015, using data from the Southern Hemisphere Influenza and Vaccine Effectiveness Research and Surveillance (SHIVERS) project. They aimed to provide estimates of RSV disease and economic burden in this cohort.

Of 4600 ARI hospitalizations tested for RSV, 348 (7.6%) were RSV positive. They found a hospitalization risk increased with: age (highest incidence among adults > 80 years), Maori or Pacific ethnicity or living in a neighborhood considered with low socioeconomic status. The average direct cost associated with each RSV hospitalization was NZD $ 4758. They conclude that an effective RSV vaccine or treatment may offer benefits for older adults.

The manuscript is well written, and easy to read, data support the conclusions, methodology is comprehensive. I will not comment the statistical analysis in detail as my knowledge in the topic is not deep.

Concerning discussion:

My first question is related to the causal attribution for RVS in patients presenting with ARI that is not discuss in the manuscript. In 47 (18%) of the cases other virus were co- detected (influenza, rhinovirus, adenovirus and human metapneumovirus). How did Authors deal with the issue of causal attribution? And what about other etiologies as such as bacterial infection? We know asymptomatic RSV infection is uncommon, but still possible. Among individuals with RSV-ARI, ARI is causally attributable to RSV in about 88% (Shi T, et al, 2019).

Concerning ARI severity (resulting in UCI admission) and co-morbidities there was no significant difference by RSV positivity status after adjusting for age and ethnicity. Any comments about this? The authors suggest a few lines below that co-morbidities may increase the risk of RSV infection in ethnic and SES populations (page 15, lines 267-8)

Due to winter seasonal incidence that approximates annual incidence on RSV infection due Authors recommend RSV screening only during winter season?

Reviewer #2: There are two issues I believe require some clarity.

The economic costs are well documented - but don't mean much to anyone outside of the NZ system and it would be helpful for readership to understand what a day in the ward costs or what the costs of a bacterial pneumonia admission would be for example.

The second and Major/critical issue is the association with Ethnicity. I understand from your description that you had adjusted ethnicity for SES - I am not happy that this is sufficient.

Unless you can provide a biological plausible reason why having Maori Genes places you at higher risk for RSV than Caucasian genes, I do not accept that you have adjusted adequately. I am not aware if there are genetic susceptibilities in Maori or pacific islanders that predispose them to viral infections?

Before concluding that "maori" is an association, I would like to see your model adjusted for SES, tobacco smoking, housing density, number of children, vaccination, nutrition, etc etc. otherwise you are merely using ethnic background as a proxy for SES which then implies that Maori has nothing to do with being Maori - but just being disadvantaged.

6. PLOS authors have the option to publish the peer review history of their article (what does this mean?). If published, this will include your full peer review and any attached files.

Reviewer #1: No

Reviewer #2: No

---

## [Author Response · Author response to Decision Letter 0]

6 May 2020

Reviewer #1: 

Comment: The authors retrospectively review a cohort of Auckland, New Zeland adult persons aged ≥18 years and identified RSV positive among ARI hospitalizations, from 2012 to 2015, using data from the Southern Hemisphere Influenza and Vaccine Effectiveness Research and Surveillance (SHIVERS) project. They aimed to provide estimates of RSV disease and economic burden in this cohort.

Of 4600 ARI hospitalizations tested for RSV, 348 (7.6%) were RSV positive. They found a hospitalization risk increased with: age (highest incidence among adults > 80 years), Maori or Pacific ethnicity or living in a neighborhood considered with low socioeconomic status. The average direct cost associated with each RSV hospitalization was NZD $ 4758. They conclude that an effective RSV vaccine or treatment may offer benefits for older adults.

The manuscript is well written, and easy to read, data support the conclusions, methodology is comprehensive. I will not comment the statistical analysis in detail as my knowledge in the topic is not deep.

Response: We thank the reviewer for their positive comments on this paper. 

Concerning discussion:

My first question is related to the causal attribution for RVS in patients presenting with ARI that is not discuss in the manuscript. In 47 (18%) of the cases other virus were co- detected (influenza, rhinovirus, adenovirus and human metapneumovirus). How did Authors deal with the issue of causal attribution? And what about other etiologies as such as bacterial infection? We know asymptomatic RSV infection is uncommon, but still possible. Among individuals with RSV-ARI, ARI is causally attributable to RSV in about 88% (Shi T, et al, 2019).

Response: Thank you for highlighting this important point. As the study reported in this manuscript estimated RSV hospitalization rates in adults by linking national administrative datasets with an ARI surveillance project, we did not have data on RSV cases among non-ARI patients. We state in the paper that our aim was to estimate ARI associated RSV hospitalisation rates and have made edits to emphasize this point throughout. 

Unfortunately we did not have systematic data on bacterial infections and could not account for this in our methods. Finally as indicated in Pg 9 line 188, not all adults were tested for all respiratory viruses. As such, we were only able to describe the type of co-infections among adults that were tested for other viruses but not account for them within our incidence estimations. We have now added a point to our discussion regarding this important limitation as shown below. 

Finally, the lack of systematic bacterial and viral co-infection data prevented detailed assessments of the causal role of RSV infection in adult ARIs. However a recent meta-analysis suggests strong evidence for this association, showing that among individuals with RSV-ARI, ARI is causally attributable to RSV in about 88% of cases.

Concerning ARI severity (resulting in UCI admission) and co-morbidities there was no significant difference by RSV positivity status after adjusting for age and ethnicity. Any comments about this? The authors suggest a few lines below that co-morbidities may increase the risk of RSV infection in ethnic and SES populations (page 15, lines 267-8)

Response: Thank you for highlighting this point. We agree this result should have been mentioned in the discussion. The lack of association between assessed comorbidities is likely driven by the fact that adults with comorbidities have a higher risk of hospitalization for non-RSV illnesses compared to those without. This is a common selection bias when carrying out analysis in hospitalized cases. We now mention this in the discussion and highlight ways in which this selection bias can be addressed as shown below

When carrying out analyses among hospitalized cases, we did not find any assessed comorbidity to be significantly associated with RSV positivity following adjustment for age and ethnicity. This lack of association is likely due to the higher risk of hospitalization for non-RSV illnesses in adults with comorbidities i.e. selection bias that occurs when both exposure and outcome are associated with hospitalization. In order to accurately estimate the effect of comorbidity on RSV disease risk, population level comorbidity data should be used to estimate RSV-associated hospitalization rates in specific co-morbidity strata. Such estimates will be valuable in identifying adult groups who are at particularly high-risk from RSV-associated disease and priority groups for future RSV vaccines and treatments.

Due to winter seasonal incidence that approximates annual incidence on RSV infection due Authors recommend RSV screening only during winter season?

Response: In Auckland, New Zealand, 90% of RSV positive nasal swab samples were detected during the winter season [Pg 5, line 97]), indicating that our seasonal rates would approximate to annual incidence. While this suggests that only seasonal surveillance for RSV could be sufficient for estimating RSV-associated disease burden – more data on RSV seasonality at the regional level and over a longer period of time is required before such a recommendation can be made. 

Reviewer #2: There are two issues I believe require some clarity.

Comment: The economic costs are well documented - but don't mean much to anyone outside of the NZ system and it would be helpful for readership to understand what a day in the ward costs or what the costs of a bacterial pneumonia admission would be for example.

Response: Thank you for this comment. We agree it would be helpful for readers to know the cost of non-RSV hospitalizations in adults. We could not find published information on average cost or bacterial pneumonia associated admission costs among adults – however we now provide additional detail on costs using the data we have – as shown below.

Based on the DRG costing methodology, the median (IQR) cost per RSV hospitalization among adults was NZD $3,723.84 ($2500.44-$5,028.44) while among non-RSV hospitalizations the median (IQR) cost per hospitalization was $3,950.71 ($2423.67-$5028.44). Hospitalization costs were not significantly different by RSV positivity in adults (p-value = 0.254).

Comment: The second and Major/critical issue is the association with Ethnicity. I understand from your description that you had adjusted ethnicity for SES - I am not happy that this is sufficient.

Unless you can provide a biological plausible reason why having Maori Genes places you at higher risk for RSV than Caucasian genes, I do not accept that you have adjusted adequately. I am not aware if there are genetic susceptibilities in Maori or pacific islanders that predispose them to viral infections?

Before concluding that "maori" is an association, I would like to see your model adjusted for SES, tobacco smoking, housing density, number of children, vaccination, nutrition, etc etc. otherwise you are merely using ethnic background as a proxy for SES which then implies that Maori has nothing to do with being Maori - but just being disadvantaged.

Response: As mentioned in the manuscript, our study used national administrative datasets linked to ARI surveillance to estimate RSV hospitalisation rates. The advantage of using such administrative datasets is that we had individual level demographic data (age, sex, ethnicity, SES status based on residential area) on all Auckland resident adults. This enabled estimation of rates by fine demographic strata. It also enabled us to carry out adjustments if necessary. We agree investigating other factors such as tobacco smoking, housing density, number of children, vaccination, and nutrition would be very valuable in better understanding RSV-associated disease risk. However, we did not have this data available at a population level and therefore could not include variables describing these exposures in our models. 

In the methods section (Pg 7, line 146) we state with references that in New Zealand, Māori and Pacific peoples have lower reported life expectancy than other ethnic groups, and are also over-represented in the lower SES groups. 

This means that ethnicity acts as a confounder on the relationship between age and RSV hospitalisation and on the relationship between SES and RSV hospitalisation and vice-versa. Thus to control for confounding and evaluate the independent effects of age, ethnicity and SES on RSV hospitalization; rates and rate ratios for age group, ethnicity, and SES were adjusted for each other.

We found that even after adjustment for age and SES, Māori and Pacific peoples still had higher rates of RSV hospitalisation compared to other ethnic groups. This finding does not necessarily mean that Māori and Pacific peoples have genetic susceptibilities that predispose them to viral infections it suggests that there are factors associated with being Māori or Pacific that increase the risk of RSV disease that are not being captured by age and area-level measure of SES. 

In the discussion (Paragraph 3) we state these higher rates of RSV hospitalisations among Māori and Pacific peoples are likely driven by differences in access to health care and health care seeking behavior as well as co-morbidities that increase the risk of RSV infection. We realize this may not have been sufficient or clear enough and have edited the paragraph as shown below. 

We observed disparities in RSV-associated hospitalization rates by age, ethnicity, and SES. Such disparities have been reported for RSV-associated hospitalization rates in children [28], but to our knowledge this is the first study to demonstrate such inequalities among adults. Ethnic differences in respiratory infectious diseases are thought primarily to be due to social and economic factors, as ethnic groups with increased risk of infection and disease{Shi, 2019 #551}se are also overrepresented in lower socio-economic groups. However in our study we found both ethnicity and SES to have independent effects on RSV-associated hospitalization risk. A potential explanation is that our neighborhood-level measure of SES may not accurately capture individual SES. Consequently, social and economic factors associated with RSV disease risk such as smoking, housing density, presence of comorbidities, health care seeking behavior, and poor nutrition are potentially being better captured by ethnicity. As our investigation is exploratory, further investigation is warranted to guide policy development. Nevertheless our findings highlight the value of assessing RSV related health disparities among different ethnic populations in other countries.

---

## [Editor Report · Decision Letter 1]

22 May 2020

The health and economic burden of respiratory syncytial virus associated hospitalizations in adults

PONE-D-19-32957R1

Dear Dr. Prasad,

We are pleased to inform you that your manuscript has been judged scientifically suitable for publication and will be formally accepted for publication once it complies with all outstanding technical requirements.

With kind regards,

Richard van Zyl-Smit

Academic Editor

PLOS ONE

Additional Editor Comments (optional):

Thank you for your detailed responses to the concerns raised and changes made to the manuscript

no further major issues
---

## [Editor Report · Acceptance letter]

29 May 2020

PONE-D-19-32957R1 

The health and economic burden of respiratory syncytial virus associated hospitalizations in adults 

Dear Dr. Prasad:

I am pleased to inform you that your manuscript has been deemed suitable for publication in PLOS ONE. Congratulations! Your manuscript is now with our production department. 

With kind regards,

on behalf of

Prof Richard van Zyl-Smit 

Academic Editor

PLOS ONE